# Association between Use of Nutritional Labeling and the Metabolic Syndrome and Its Components

**DOI:** 10.3390/ijerph16224486

**Published:** 2019-11-14

**Authors:** Hyung-sub Jin, Eun-bee Choi, Minseo Kim, Sarah Soyeon Oh, Sung-In Jang

**Affiliations:** 1Medical Courses, Yonsei University College of Medicine, Seoul 03722, Korea; 2Institute of Health Services Research, Yonsei University, Seoul 03722, Korea; 3Department of Public Health, Graduate School, Yonsei University, Seoul 03722, Korea; 4Department of Preventive Medicine, Yonsei University College of Medicine, Seoul 03722, Korea

**Keywords:** metabolic syndrome, nutritional labeling, Korea National Health and Nutritional Examination Survey, smoking, drinking, stress

## Abstract

In this study, we looked into the association between the diagnosis of metabolic syndrome (MetS) and nutritional label awareness. This study used data from the Korea National Health and Nutritional Examination Survey (KNHANES) for the years 2007 to 2015. The study population consisted of a total of 41,667 Koreans of which 11,401 (27.4%) were diagnosed with metabolic syndrome and 30,266 (72.6%) were not. Groups not using nutritional labeling had a 24% increase in odds risk (OR: 1.24, 95% CI 1.14–1.35) of MetS compared to groups using nutritional labeling. Use of nutritional labeling was associated with all components of MetS. Central obesity showed the highest increase in odds risk (OR: 1.23, 95% CI 1.13–1.35) and high blood pressure showed the lowest increase in odds risk (OR: 1.11, 95% CI 1.02–1.20). Subgroup analysis revealed that statistically significant factors were smoking status, drinking status and stress status. Groups that smoke, groups that do not drink and groups with high stress were more vulnerable to MetS when not using nutritional labeling. People not using food labels tends to develop metabolic syndromes more than people using foods labels. In the subgroup analysis, drinking status, smoking status and stress status were significant factors.

## 1. Introduction

According to a study by the Korean Statistical Information Service (KOSIS) in 2017, cardiovascular diseases and diabetes were ranked inside the top ten causes of death in Korea [1].

It is widely known that people with metabolic syndrome, which is not a specific disease but a cluster of attributes, including hyperglycemia, insulin resistance, hypertension, and raised VLDL-triglycerides [2], have a higher probability of developing cardiovascular disease and diabetes mellitus, with higher mortality from all causes as well as cardiovascular disease [3,4,5].

Contracting metabolic syndromes is known to approximately double the risk of cardiovascular disease and quintuple the risk in type 2 diabetes over 5 to 10 years [6]. Therefore, in an attempt to lower the number of deaths caused by these two high-mortality diseases, researches about metabolic syndromes and lifestyle are rapidly being conducted.

Numerous studies suggest a positive correlation between alcohol consumption and metabolic syndrome [7,8,9,10]. Most studies analyzing the occurrence of metabolic syndrome and characteristics of humans cover lifestyle and genetic characteristics, such as ethnicity, gender, age, diabetes and obesity [11,12,13,14,15]. However, there are a limited number of researches about the association between metabolic syndrome and nutrition label awareness. The number of studies that relate metabolic syndrome and nutrition label comprehension deal with the U.S. population, not the Korean population [16]. Furthermore, researches analyzing this association using credible research data covering more than five years are not prevalent, with most studies only handling two to three years [17,18]. With a need to analyze the Korean population within a longer time period, our team decided to look into the metabolic syndrome occurrence and nutrition labeling comprehension of the Korean population from 2007 to 2015, a total of nine years.

In this study, we tried to show a relationship between use of nutritional labeling and metabolic syndrome (MetS). We analyzed the association between metabolic syndrome and nutrition label awareness, as well as sex, age, degree of physical activity, occupied area, smoking status, income, occupation and academic level within the year 2007 to 2015. In an effort to resolve the problem that metabolic syndrome is not a clear disease, we used five standards (central obesity, high triglycerides, low HDL cholesterol, high blood pressure and high fasting plasma glucose) to determine the condition’s presence.

## 2. Materials and Methods

### 2.1. Study Population and Data

This study was conducted using data from the Korea National Health and Nutrition Examination Survey (KNHANES). KHANES gives statistic information about the health and nutritional status of the population and select the health-vulnerable groups that need to be prioritized. The survey also provides statistics for health-related policies in Korea, which also serve as the research infrastructure for studies on risk factors and diseases by supporting over 500 publications [19].

The target population of KNHANES comprises non-institutionalized Korean citizens residing in Korea. The sampling plan follows a multi-stage clustered probability design. For example, in the 2011 survey, 192 primary sampling units (PSUs) were drawn from approximately 200,000 geographically defined PSUs for the whole country. A PSU consisted of an average of 60 households, and 20 final target households were sampled for each PSU using systematic sampling; in the selected households, individuals aged 1 year and over were targeted. The number of participants is shown in Table 1. The numbers of participants of the first three surveys (1998, 2001 and 2005) were approximately 35,000 in each survey. From 2007 the survey became a continuous programme with about 10,000 individuals each year except for the year 2007, when the number of participants was half of that of other years as the 2007 survey was conducted during a half-year (from July through December). All the statistics of this survey were calculated using sample weights assigned to sample participants.

The KNHANES is a national surveillance system that has been assessing the health and nutritional status of Koreans since 1998. The survey is based on the National Health Promotion Act, and the surveys have been conducted by the Korea Centers for Disease Control and Prevention (KCDC). Approximately 10,000 individuals were selected from 192 primary sampling units (PSUs) around the country [19].

### 2.2. Variables

In this study, metabolic syndrome (MetS) and its components was selected as the outcome variable. The presence of MetS was measured using the guidelines provided by the Korean Academy of Medical Sciences. According to the Korean Academy of Medical Sciences those with MetS have three of the following five features: (1) centrally obese (measured by a waist circumference of ≥90 cm if male and ≥80 cm if female); (2) an increased triglyceride level of ≥150 mg/dL; (3) a decreased high density lipoprotein cholesterol level of <40 mg/dL in men and <50 mg/dL in women; (4) raised blood pressure, indicated by a systolic blood pressure of ≥130 mmHg, or a diastolic blood pressure of ≥85 mmHg, or treatment of previously diagnosed hypertension; and (5) an increased fasting plasma glucose level of ≥100 mg/dL. Such components, as well as all health-related components of the KNHANES, were collected via standardized physical examination by medical technicians serving as staff members for the survey.

Use of nutritional labeling when choosing the food was surveyed by KNHANES and was categorized into the following two groups: (1) Yes and (2) No.

Various demographic, socioeconomic and health-related covariates were included. Covariates included sex (male, female), age (20–29, 30–39, 40–49, 50–59, 60–69, 70–79, ≥80), region (urban, rural), household income group (low, medium-low, medium-high, high), occupation (white collar, sales and services, blue collar), educational attainment (≤elementary school, middle school, high school diploma, ≥bachelor’s degree), obesity (underweight, normal weight, overweight), smoking status (non-smoker, smoker), drinking status (non-drinker, drinker) and stress status (low stress, high stress). Income groups were obtained by dividing household income by the square root of the number of household members and divided it into four groups using quartiles. These variables are profound factors for MetS, and we controlled these variables in our study.

### 2.3. Statistical Analysis

To examine the association between the use of nutritional labeling and MetS and its components, multiple logistic regression analysis was performed using the data. Odds ratios and 95% confidence intervals (CIs) were calculated to compare between the using nutritional labeling group and the non-using nutritional labeling group.

Our study population consisted of 19,368 Korean males and 22,299 Korean females over 20 years of age from 2007 to 2015. There were no missing subjects from the initial population. All analyses were performed using SAS software, version 9.4 (SAS Institute, Cary, NC, USA).

## 3. Results

### 3.1. Study Participants

Table 1 and Table 2 present the results for the general characteristics of the 41,667 Koreans above the age of 20, from 2007 to 2015, within our final study population. A total of 11,401 (27.4%) were diagnosed with MetS and 30,266 (72.6%) were not. A total of 5907 (14.2%) used nutritional labeling when choosing the food and 35,760 (85.8%) did not.

Among the 5907 people who used nutritional labeling, 1043 (17.7%) were diagnosed with MetS and 4864 (82.3%) were not. For the 35,760 people who also did not use nutritional labeling 10,358 (29.0%) were diagnosed with MetS and 25,402 (71.0%) were not.

### 3.2. Relationship between MetS and Use of Nutritional Labeling

Table 3 presents the results of multiple logistic regression analysis of the study population for MetS adjusted for the following variables: use of nutritional labeling, sex, age, region, household income, occupation, educational attainment, obesity, smoking status, drinking status and stress status. Table 4 presents the results of multiple logistic regression analysis of the study population for casual components of MetS adjusted for the same variables in Table 3.

Groups not using the nutritional labeling had a 24% increase in odds risk (OR: 1.24, 95% CI 1.14–1.35) of MetS compared to groups using the nutritional labeling. As the people got older, the odds risk of MetS also increased. Groups that smoke (OR: 1.36, 95% CI 1.27–1.46) and groups with high stress (OR: 1.10, 95% CI 1.04–1.17) also had a higher odds risk of MetS (Table 3). Groups not using the nutritional labeling showed an increase in odds risk compared to groups using the nutritional labeling for the five components of MetS. Central obesity showed the highest increase in odds risk (OR: 1.23, 95% CI 1.13–1.35) and high blood pressure showed the lowest increase in odds risk (OR: 1.11, 95% CI 1.02–1.20) when not using the nutritional labeling (Table 4).

### 3.3. Subgroup Analysis

Table 5 and Table 6 presents the subgroup analysis of the study population. Performing subgroup analysis, the statistically significant factors were smoking status, drinking status and stress status. Groups that smoke (OR: 1.39, 95% CI 1.11–1.75) are more vulnerable to MetS when not using the nutritional labeling compared to non-smoking groups (OR: 1.17, 95% CI 1.07–1.29) (Table 5). Non-drinking groups (OR: 1.23, 95% CI 1.10–1.39) are more vulnerable to MetS than drinking groups (OR: 1.19, 95% CI 1.04–1.36) when not using the nutritional labeling. Finally, high stress groups (OR: 1.32, 95% CI 1.11–1.56) are more vulnerable to MetS then low stress groups (OR: 1.21, 95% CI 1.09–1.34) when not using the nutritional labeling.

## 4. Discussion

We found that the use of nutritional labeling is associated with metabolic syndrome across the whole observation. We also found that the use of nutritional labeling is associated with decreased metabolic syndrome in the subgroups divided by smoking status, drinking status and stress status. However, in most of the groups divided by other variations, there was no consistent effect of the use of nutritional labeling on metabolic syndrome.

There are numerous previous studies regarding the use of food labels among adults with metabolic syndrome. One study shows that patients with metabolic syndrome tends to use food labels less than adults with no metabolic syndrome [20]. This issue is noteworthy because diet is one of the important ways to treat metabolic syndrome. Especially diets limiting intake of saturated fat and with high fiber/low glycemic-index is an effective treatment for metabolic syndrome [21]. Through our research about the relationship between metabolic syndrome and the use of food labeling, the signs are that the use of food labeling is not only necessary for patients with metabolic syndrome but also not using food labels might be one of the causes of metabolic syndrome because diet and metabolic syndrome are closely related. Especially there are significant association between intake of fat and cholesterol and metabolic syndrome in men and intake of carbohydrate and metabolic syndrome in women [22]. In addition, there was research showing that the use of food labels affects intake of nutrients, including total fat, total energy, saturated fat, cholesterol, sodium, dietary fiber and sugars, in a healthier way in the US [16].

There are several limitations of this research. First, there might be other confounders that must be considered because the use of labels was found to be associated with several factors, such as sex, age and socioeconomic status [20]. Therefore, the use of food labels might not be a direct cause of metabolic syndrome and the odds ratio of people not using food labels developing metabolic syndrome might be overrated. It is also impossible to measure the effects of using food labels on developing metabolic syndrome exactly. Even though we have data suggesting that people using food labels are less likely to develop metabolic syndrome, there would be some people who stopped reading food labels or started using food labels after being diagnosed with metabolic syndrome. If there are some people who started using food labels after being diagnosed with metabolic syndrome in the data, then the effects of using food labels on developing metabolic syndrome might be greater than we can infer from this research.

We concluded that the use of nutritional labeling has a significant association with metabolic syndrome with a 1.24 odds ratio and 1.14–1.35 95% CI. Although there are some groups with no consistent association between these two factors in the subgroup analysis, in the groups divided by smoking status, drinking status and stress status there was significant association between these two factors. If there are more detailed life trajectory data of using food labels and being diagnosed with metabolic syndrome, then it would be possible to find out more about the relationship between these two factors. Even though we had some limitations with our method, this research still supports the association between the use of food labels and metabolic syndrome. Especially, it shows the odds ratio in each feature of metabolic syndrome and they are all significant in the whole study population. It also shows the odds ratio in each feature in the subgroup analysis. This can be helpful to figure out the way the use of food labels affects metabolic syndrome. This research emphasizes the importance of diet in preventing and treating metabolic syndrome.

## 5. Conclusions

We found out that people not using food labels tend to develop metabolic syndrome more than people using foods labels. Furthermore, people with a positive drinking status, smoking status or stress status were more vulnerable to metabolic syndromes when not using food labels. When discussing MetS, the type of nutrition should also be considered as a prime factor. So, we were working under the assumption that the group using nutritional labeling tend to show more concern for the type of nutrition on their diet. By that assumption we could just work on showing a relationship between use of nutritional labeling and MetS. Further studies are needed to show that there is a relationship between using nutritional labeling and the type of nutrition which they take in. But we can still say that by using nutritional labeling we can decrease the probability of MetS. Therefore, we suggest that to prevent metabolic syndrome, education regarding using food labels are recommended, especially for people who drink, smoke or have stress.

## Figures and Tables

**Table 1 ijerph-16-04486-t001:** General characteristics of the study observations (2007–2015).

	Metabolic Syndrome					
	Total		Yes		No	
	*n*	(%)	*n*	(%)	*n*	(%)
**Use of Nutritional Labeling**						
Yes	5907	(14.2)	1043	(17.7)	4864	(82.3)
No	35,760	(85.8)	10,358	(29.0)	25,402	(71.0)
**Sex**						
Male	19,368	(46.5)	5727	(29.6)	13,641	(70.4)
Female	22,299	(53.5)	5674	(25.5)	16,625	(74.6)
**Age**						
20–29	5336	(12.8)	329	(6.2)	5007	(93.8)
30–39	8642	(20.7)	1187	(13.7)	7455	(86.3)
40–49	8416	(20.2)	2003	(23.8)	6413	(76.2)
50–59	7791	(18.7)	2729	(35.0)	5062	(65.0)
60–69	6467	(15.5)	2918	(45.1)	3549	(54.9)
70–79	4211	(10.1)	1919	(45.6)	2292	(54.4)
≥80	804	(1.9)	316	(39.3)	488	(60.7)
**Region**						
Urban	16,699	(40.1)	4408	(26.4)	12,291	(73.6)
Rural	24,968	(59.9)	6993	(28.0)	17,975	(72.0)
**Household Income**						
Low	7292	(17.5)	2860	(39.2)	4432	(60.8)
Medium-low	10,492	(25.2)	3042	(29.0)	7450	(71.0)
Medium-high	11,747	(28.2)	2816	(24.0)	8931	(76.0)
High	12,136	(29.1)	2683	(22.1)	9453	(77.9)
**Occupation**						
White Collar	14,808	(35.5)	3257	(22.0)	11,551	(78.0)
Sales and Services	10,890	(26.1)	3317	(30.5)	7573	(69.5)
Blue Collar	15,969	(38.3)	4827	(30.2)	11,142	(69.8)
**Educational Attainment**						
≤Elementary School	9111	(21.9)	4141	(45.5)	4970	(54.6)
Middle School	4539	(10.9)	1597	(35.2)	2942	(64.8)
High School Diploma	14,675	(35.2)	3324	(22.7)	11,351	(77.4)
≥Bachelor’s Degree	13,342	(32.0)	2339	(17.5)	11,003	(82.5)
**Obesity**						
Underweight	1882	(4.5)	40	(2.1)	1842	(97.9)
Normal weight	26,583	(63.8)	4304	(16.2)	22,279	(83.8)
Overweight	13,202	(31.7)	7057	(53.5)	6145	(46.6)
**Smoking Status**						
Non-smoker	32,825	(78.8)	8891	(27.1)	23,934	(72.9)
Smoker	8842	(21.2)	2510	(28.4)	6332	(71.6)
**Drinking Status**						
Non-drinker	16,281	(30.1)	4680	(28.8)	11,601	(71.3)
Drinker	25,386	(60.9)	6721	(26.5)	18,665	(73.5)
**Stress Status**						
Low stress	30,427	(73.0)	8417	(27.7)	22,010	(72.3)
High stress	11,240	(27.0)	2984	(26.6)	8256	(73.5)
**Year**						
2007	2267	(5.4)	698	(30.8)	1569	(69.2)
2008	5493	(13.2)	1365	(24.9)	4128	(75.2)
2009	6151	(14.8)	1532	(24.9)	4619	(75.1)
2010	5120	(12.3)	1258	(24.6)	3862	(75.4)
2011	5032	(12.1)	1246	(24.8)	3786	(75.2)
2012	4621	(11.1)	1203	(26.0)	3418	(74.0)
2013	4502	(10.8)	1140	(25.3)	3362	(74.7)
2014	4208	(10.1)	1101	(26.2)	3107	(73.8)
2015	4273	(10.3)	1858	(43.5)	2415	(56.5)
**Total**	41,667	(100.0)	11,401	(100.0)	30,266	(100.0)

**Table 2 ijerph-16-04486-t002:** General Characteristics of Study Observations of Metabolic Syndrome’s components (2007–2015).

	Central Obesity						High Triglycerides						Low HDL Cholesterol						High Blood Pressure						High Fasting Plasma Glucose					
	Total		Yes		No		Total		Yes		No		Total		Yes		No		Total		Yes		No		Total		Yes		No	
	*n*	(%)	*n*	(%)	*n*	(%)	*n*	(%)	*n*	(%)	*n*	(%)	*n*	(%)	*n*	(%)	*n*	(%)	*n*	(%)	*n*	(%)	*n*	(%)	*n*	(%)	*n*	(%)	*n*	(%)
**Use of Nutritional Labeling**																														
Yes	5907	14.2	1704	28.9	4203	71.2	5907	14.2	1144	19.4	4763	80.6	5907	14.2	2314	39.2	3593	60.8	5907	14.2	1336	22.6	4571	77.4	5907	14.2	1061	18.0	4846	82.0
No	35,760	85.8	12,701	35.5	23,059	64.5	35,760	85.8	10,352	29.0	25,408	71.1	35,760	85.8	14,405	40.3	21,355	59.7	35,760	85.8	14,399	40.3	21,361	59.7	35,760	85.8	10,416	29.1	25,344	70.9
**Sex**																														
Male	19,368	46.5	5260	27.2	14,108	72.8	19,368	46.5	7069	36.5	12,299	63.5	19,368	46.5	6502	33.6	12,866	66.4	19,368	46.5	8983	46.4	10,385	53.6	19,368	46.5	6719	34.7	12,649	65.3
Female	22,299	53.5	9145	41.0	13,154	59.0	22,299	53.5	4427	19.9	17,872	80.2	22,299	53.5	10,217	45.8	12,082	54.2	22,299	53.5	6752	30.3	15,547	69.7	22,299	53.5	4758	21.3	17,541	78.7
**Age**																														
20–29	5336	12.8	920	17.2	4416	82.8	5336	12.8	722	15.5	4614	84.5	5336	12.8	1654	31.0	3682	69.0	5336	12.8	522	9.8	4814	90.2	5336	12.8	305	5.7	5,031	94.3
30–39	8642	20.7	2277	26.4	6365	73.7	8642	20.7	1963	22.7	6679	77.3	8642	20.7	3042	35.2	5600	64.8	8642	20.7	1360	15.7	7282	84.3	8642	20.7	1257	14.6	7385	85.5
40–49	8416	20.2	2661	31.6	5755	68.4	8416	20.2	2525	30.0	5891	70.0	8416	20.2	3226	38.3	5190	61.7	8416	20.2	2533	30.1	5883	69.9	8416	20.2	2192	26.1	6224	74.0
50–59	7791	18.7	364	40.6	4627	59.4	7791	18.7	2692	34.6	5099	65.5	7791	18.7	3175	40.8	4616	59.3	7791	18.7	3697	47.5	4094	52.6	7791	18.7	2922	37.5	4869	62.5
60–69	6467	15.5	3106	48.0	3361	52.0	6467	15.5	2221	34.3	4246	65.7	6467	15.5	3002	46.4	3465	53.6	6467	15.5	4033	62.4	2434	37.6	6467	15.5	2860	44.2	3607	55.8
70–79	4211	10.1	1973	46.9	2238	53.2	4211	10.1	1219	29.0	2992	71.1	4211	10.1	2085	49.5	2126	50.5	4211	10.1	2990	71.0	1221	29.0	4211	10.1	1710	40.6	2501	59.4
≥80	804	1.9	304	37.8	500	62.2	804	1.9	154	19.2	650	80.9	804	1.9	535	66.5	269	33.5	804	1.9	600	74.6	204	25.4	804	1.9	231	28.7	573	71.3
**Region**																														
Urban	16,699	40.1	5469	32.8	11,230	67.3	16,699	40.1	4462	26.7	12,237	73.3	16,699	40.1	6484	38.8	10,215	61.2	16,699	40.1	6183	37.0	10,516	63.0	16,699	40.1	4420	26.5	12,279	73.5
Rural	24,968	59.9	8936	35.8	16,032	64.2	24,968	59.9	7034	28.2	17,934	71.8	24,968	59.9	10,235	41.0	14,733	59.0	24,968	59.9	9552	38.3	15,416	61.7	24,968	59.9	7057	28.3	17,911	71.7
**Household Income**																														
Low	7292	17.5	3175	43.5	4117	56.5	7292	17.5	2227	30.5	5065	69.5	7292	17.5	3477	47.7	3815	52.3	7292	17.5	4149	56.9	3143	43.1	7292	17.5	2530	34.7	4762	65.3
Medium-low	10,492	25.2	3890	37.1	6602	62.9	10,492	25.2	2919	27.8	7573	72.2	10,492	25.2	4274	40.7	6218	59.3	10,492	25.2	4110	39.2	6382	60.8	10,492	25.2	3009	28.7	7483	71.3
Medium-high	11,747	28.2	3789	32.3	7958	67.7	11,747	28.2	3178	27.1	8569	73.0	11,747	28.2	4497	38.3	7250	61.7	11,747	28.2	3825	32.6	7922	67.4	11,747	28.2	2982	25.4	8765	74.6
High	12,136	29.1	3551	29.3	8585	70.7	12,136	29.1	3172	26.1	8964	73.9	12,136	29.1	4471	36.8	7665	63.2	12,136	29.1	3651	30.1	8485	69.9	12,136	29.1	2956	24.4	9180	75.6
**Occupation**																														
White Collar	14,808	35.5	4349	29.4	10,459	70.6	14,808	35.5	4058	27.4	10,750	72.6	14,808	35.5	5405	36.5	9403	63.5	14,808	35.5	4198	28.4	10,610	71.7	14,808	35.5	3448	23.3	11,360	76.7
Sales and Services	10,890	26.1	3786	34.8	7104	65.2	10,890	26.1	3466	31.8	7424	68.2	10,890	26.1	3981	36.6	6909	63.4	10,890	26.1	5042	46.3	5848	53.7	10,890	26.1	3727	34.2	7163	65.8
Blue Collar	15,969	38.3	6270	39.3	9699	60.7	15,969	38.3	3972	24.9	11,997	75.1	15,969	38.3	7333	45.9	8636	54.1	15,969	38.3	6495	40.7	9474	59.3	15,969	38.3	4302	26.9	11,667	73.1
**Educational Attainment**																														
≤Elementary School	9111	21.9	4708	51.7	4403	48.3	9111	21.9	2992	32.8	6119	67.2	9111	21.9	4562	50.1	4549	49.9	9111	21.9	5758	63.2	3353	36.8	9111	21.9	3543	38.9	5568	61.1
Middle School	4539	10.9	1872	41.2	2667	58.8	4539	10.9	1458	32.1	3081	67.9	4539	10.9	1942	42.8	2597	57.2	4539	10.9	2223	49.0	2316	51.0	4539	10.9	1676	36.9	2863	63.1
High School Diploma	14,675	35.2	4470	30.5	10,205	69.5	14,675	35.2	3812	26.0	10,863	74.0	14,675	35.2	5520	37.6	9155	62.4	14,675	35.2	4532	30.9	10,143	69.1	14,675	35.2	3644	24.8	11,031	75.2
≥Bachelor’s Degree	13,342	32.0	3355	25.2	9987	74.9	13,342	32.0	3234	24.2	10,108	75.8	13,342	32.0	4695	35.2	8647	64.8	13,342	32.0	3222	24.2	10,120	75.9	13,342	32.0	2614	19.6	10,728	80.4
**Obesity**																														
Underweight	1882	4.5	15	0.8	1867	99.2	1882	4.5	107	5.7	1775	94.3	1882	4.5	552	29.3	1330	70.7	1882	4.5	284	15.1	1598	84.9	1882	4.5	160	8.5	1722	91.5
Normal weight	26,583	63.8	4320	16.3	22,263	83.8	26,583	63.8	5784	21.8	20,799	78.2	26,583	63.8	9767	36.7	16,816	63.3	26,583	63.8	8416	31.7	18,167	68.3	26,583	63.8	5973	22.5	20,610	77.5
Overweight	13,202	31.7	10,070	76.3	3132	23.7	13,202	31.7	5605	42.5	7597	57.5	13,202	31.7	6400	48.5	6802	51.5	13,202	31.7	7035	53.3	6167	46.7	13,202	31.7	5344	40.5	7858	59.5
**Smoking Status**																														
Non-smoker	32,825	78.8	11,940	36.4	20,885	63.6	32,825	78.8	7935	24.2	24,890	75.8	32,825	78.8	13,679	41.7	19,146	58.3	32,825	78.8	12,285	37.4	20,540	62.6	32,825	78.8	8799	26.8	24,026	73.2
Smoker	8842	21.2	2465	27.9	6377	72.1	8842	21.2	3561	40.3	5281	59.7	8842	21.2	3040	34.4	5802	65.6	8842	21.2	3450	39.0	5392	61.0	8842	21.2	2678	30.3	6164	69.7
**Drinking Status**																														
Non-drinker	16,281	30.1	6377	39.2	9904	60.8	16,281	30.1	3822	23.5	12,459	76.5	16,281	30.1	7922	48.7	8359	51.3	16,281	30.1	6041	37.1	10,240	62.9	16,281	30.1	4135	25.4	12,146	74.6
Drinker	25,386	60.9	8028	31.6	17,358	68.4	25,386	60.9	7674	30.2	17,712	69.8	25,386	60.9	8797	34.7	16,589	65.4	25,386	60.9	9694	38.2	15,692	61.8	25,386	60.9	7342	28.9	18,044	71.1
**Stress Status**																														
Low stress	30,427	73.0	10,442	34.3	19,985	65.7	30,427	73.0	8382	27.6	22,045	72.5	30,427	73.0	12,146	39.9	18,281	60.1	30,427	73.0	11,876	39.0	18,551	61.0	30,427	73.0	8646	28.4	21,781	71.6
High stress	11,240	27.0	3963	35.3	7277	64.7	11,240	27.0	3114	27.7	8126	72.3	11,240	27.0	4573	40.7	6667	59.3	11,240	27.0	3859	34.3	7381	65.7	11,240	27.0	2831	25.2	8409	74.8
**Year**																														
2007	2267	5.4	867	38.2	1400	61.8	2267	5.4	677	29.9	1590	70.1	2267	5.4	1465	64.6	802	35.4	2267	5.4	717	31.6	1,550	68.4	2267	5.4	535	23.6	1732	76.4
2008	5493	13.2	2010	36.6	3483	63.4	5493	13.2	1537	28.0	3956	72.0	5493	13.2	1786	32.5	3707	67.5	5493	13.2	1778	32.4	3715	67.6	5493	13.2	1510	27.5	3983	72.5
2009	6151	14.8	2074	33.7	4077	66.3	6151	14.8	1697	27.6	4454	72.4	6151	14.8	1908	31.0	4243	69.0	6151	14.8	2449	39.8	3702	60.2	6151	14.8	1567	25.5	4584	74.5
2010	5120	12.3	1637	32.0	3483	68.0	5120	12.3	1374	26.8	3746	73.2	5120	12.3	1532	29.9	3588	70.1	5120	12.3	2100	41.0	3020	59.0	5120	12.3	1261	24.6	3859	75.4
2011	5032	12.1	1765	35.1	3267	64.9	5032	12.1	1402	27.9	3630	72.1	5032	12.1	1401	27.8	3631	72.2	5032	12.1	1974	39.2	3058	60.8	5032	12.1	1327	26.4	3705	73.6
2012	4621	11.1	1560	33.8	3061	66.2	4621	11.1	1245	26.9	3376	73.1	4621	11.1	1552	33.6	3069	66.4	4621	11.1	1804	39.0	2817	61.0	4621	11.1	1292	28.0	3329	72.0
2013	4502	10.8	1395	31.0	3107	69.0	4502	10.8	1212	26.9	3290	73.1	4502	10.8	1458	32.4	3044	67.6	4502	10.8	1645	36.5	2857	63.5	4502	10.8	1342	29.8	3160	70.2
2014	4208	10.1	1418	33.7	2790	66.3	4208	10.1	1160	27.6	3048	72.4	4208	10.1	1344	31.9	2864	68.1	4208	10.1	1554	36.9	2654	63.1	4208	10.1	1229	29.2	2979	70.8
2015	4273	10.3	1679	39.3	2594	60.7	4273	10.3	1192	27.9	3081	72.1	4273	10.3	4273	100.0	0	0.0	4273	10.3	1714	40.1	2559	59.9	4273	10.3	1414	33.1	2859	66.9
**Total**	41,667	100.0	14,405	100.0	27,262	100.0	41,667	100.0	11,496	100.0	30,171	100.0	41,667	100.0	16,719	100.0	24,948	100.0	41,667	100.0	15,735	100.0	25,932	100.0	41,667	100.0	11,477	100.0	30,190	100.0

**Table 3 ijerph-16-04486-t003:** Factors associated with metabolic syndrome (2007–2015).

	Metabolic Syndrome			
	Odds Ratio	95% CI *		
**Use of Nutritional Labeling**				
Yes	1.00	-		-
No	**1.24**	**(1.14**	–	**1.35)**
**Sex**				
Male	1.00	-		-
Female	0.98	(0.92	–	1.04)
**Age**				
20–29	1.00	-		-
30–39	**2.36**	**(2.07**	–	**2.71)**
40–49	**4.41**	**(3.86**	–	**5.02)**
50–59	**7.00**	**(6.13**	–	**8.00)**
60–69	**10.44**	**(9.08**	–	**12.00)**
70–79	**11.27**	**(9.69**	–	**13.11)**
≥80	**10.21**	**(8.26**	–	**12.61)**
**Region**				
Urban	1.00	-		-
Rural	1.00	(0.95	–	1.05)
**Household Income**				
Low	1.00	-		-
Medium-low	0.95	(0.88	–	1.03)
Medium-high	0.93	(0.85	–	1.01)
High	**0.88**	**(0.81**	–	**0.96)**
**Occupation**				
White Collar	1.00	-		-
Sales and Services	0.80	(0.74	–	0.86)
Blue Collar	**1.07**	**(0.99**	–	**1.14)**
**Educational Attainment**				
≤Elementary School	1.00	-		-
Middle School	**0.77**	**(0.70**	–	**0.84)**
High School Diploma	**0.71**	**(0.65**	–	**0.77)**
≥Bachelor’s Degree	**0.60**	**(0.54**	–	**0.66)**
**Obesity**				
Underweight	**0.13**	**(0.10**	–	**0.18)**
Normal weight	1.00	-		-
Overweight	**6.73**	**(6.39**	–	**7.09)**
**Smoking Status**				
Non-smoker	1.00	-		-
Smoker	**1.36**	**(1.27**	–	**1.46)**
**Drinking Status**				
Non-drinker	1.00	-		-
Drinker	1.05	(0.99	–	1.11)
**Stress Status**				
Low stress	1.00	-		-
High stress	**1.10**	**(1.04**	–	**1.17)**
**Year**				
2007	1.00	-		-
2008	**0.70**	**(0.62**	–	**0.80)**
2009	**0.70**	**(0.62**	–	**0.79)**
2010	**0.71**	**(0.62**	–	**0.81)**
2011	**0.66**	**(0.58**	–	**0.76)**
2012	**0.72**	**(0.63**	–	**0.82)**
2013	**0.72**	**(0.63**	–	**0.82)**
2014	**0.74**	**(0.65**	–	**0.84)**
2015	**1.98**	**(1.74**	–	**2.25)**

* CI: Confidence interval. The bolds here are to show that they are the significant variables.

**Table 4 ijerph-16-04486-t004:** Factors associated with the casual factors of metabolic syndrome’s components (2007–2015).

	Central Obesity				High Triglycerides				Low HDL Cholesterol				High Blood Pressure				High Fasting Plasma Glucose			
	Odds Ratio	95% CI *			Odds Ratio	95% CI *			Odds Ratio	95% CI *			Odds Ratio	95% CI *			Odds Ratio	95% CI *		
**Use of Nutritional Labeling**																				
Yes	1.00	-		-	1.00	-		-	1.00	-		-	1.00	-		-	1.00	-		-
No	**1.23**	**(1.13**	–	**1.35)**	**1.15**	**(1.07**	–	**1.25)**	**1.21**	**(1.12**	–	**1.30)**	**1.11**	**(1.02**	–	**1.20)**	**1.13**	**(1.04**	–	**1.22)**
**Sex**																				
Male	1.00	-		-	1.00	-		-	1.00	-		-	1.00	-		-	1.00	-		-
Female	**5.06**	**(4.69**	–	**5.45)**	**0.52**	**(0.49**	–	**0.55)**	**1.95**	**(1.84**	–	**2.06)**	**0.51**	**(0.48**	–	**0.54)**	**0.56**	**(0.53**	–	**0.59)**
**Age**																				
20–29	1.00	-		-	1.00	-		-	1.00	-		-	1.00	-		-	1.00	-		-
30–39	**1.56**	**(1.40**	–	**1.74)**	**1.82**	**(1.65**	–	**2.00)**	**1.31**	**(1.20**	–	**1.43)**	**1.70**	**(1.52**	–	**1.90)**	**2.80**	**(2.45**	–	**3.20)**
40–49	**1.81**	**(1.62**	–	**2.02)**	**2.53**	**(2.29**	–	**2.78)**	**1.41**	**(1.29**	–	**1.54)**	**3.69**	**(3.32**	–	**4.11)**	**5.39**	**(4.74**	–	**6.13)**
50–59	**2.64**	**(2.36**	–	**2.97)**	**2.87**	**(2.59**	–	**3.17)**	**1.35**	**(1.23**	–	**1.48)**	**7.07**	**(6.34**	–	**7.88)**	**8.67**	**(7.61**	–	**9.87)**
60–69	**4.30**	**(3.78**	–	**4.88)**	**2.57**	**(2.31**	–	**2.87)**	**1.67**	**(1.51**	–	**1.85)**	**11.57**	**(10.31**	–	**12.98)**	**11.01**	**(9.61**	–	**12.60)**
70–79	**4.95**	**(4.28**	–	**5.73)**	**1.96**	**(1.74**	–	**2.22)**	**1.94**	**(1.73**	–	**2.18)**	**16.90**	**(14.86**	–	**19.23)**	**9.72**	**(8.41**	–	**11.24)**
≥80	**3.89**	**(3.10**	–	**4.87)**	**1.25**	**(1.01**	–	**1.54)**	**4.42**	**(3.66**	–	**5.34)**	**22.94**	**(18.79**	–	**28.02)**	**6.35**	**(5.16**	–	**7.81)**
**Region**																				
Urban	1.00	-		-	1.00	-		-	1.00	-		-	1.00	-		-	1.00	-		-
Rural	**1.11**	**(1.05**	–	**1.18)**	**1.04**	**(0.99**	–	**1.09)**	**1.08**	**(1.03**	–	**1.13)**	**0.92**	**(0.88**	–	**0.97)**	**1.02**	**(0.98**	–	**1.08)**
**Household Income**																				
Low	1.00	-		-	1.00	-		-	1.00	-		-	1.00	-		-	1.00	-		-
Medium-low	1.02	(0.93	–	1.11)	0.93	(0.87	–	1.00)	0.95	(0.88	–	1.02)	0.91	(0.85	–	0.98)	1.02	(0.94	–	1.09)
Medium-high	0.97	(0.88	–	1.06)	0.96	(0.89	–	1.04)	0.94	(0.87	–	1.01)	0.91	(0.84	–	0.98)	1.03	(0.95	–	1.11)
High	0.93	(0.84	–	1.02)	0.94	(0.87	–	1.02)	0.89	(0.82	–	0.96)	0.83	(0.77	–	0.90)	0.99	(0.91	–	1.07)
**Occupation**																				
White Collar	1.00	-		-	1.00	-		-	1.00	-		-	1.00	-		-	1.00	-		-
Sales and Services	**0.89**	**(0.83**	–	**0.97)**	**0.84**	**(0.78**	–	**0.89)**	**0.83**	**(0.78**	–	**0.89)**	**0.94**	**(0.88**	–	**1.00)**	**0.94**	**(0.88**	–	**1.00)**
Blue Collar	**1.12**	**(1.05**	–	**1.21)**	**1.04**	**(0.97**	–	**1.10)**	**1.08**	**(1.02**	–	**1.15)**	**1.05**	**(0.98**	–	**1.11)**	**1.01**	**(0.94**	–	**1.08)**
**Educational Attainment**																				
≤Elementary School	1.00	-		-	1.00	-		-	1.00	-		-	1.00	-		-	1.00	-		-
Middle School	**0.80**	**(0.72**	–	**0.88)**	**0.81**	**(0.74**	**-**	**0.88)**	**0.96**	**(0.88**	–	**1.05)**	**0.75**	**(0.69**	–	**0.81)**	**0.95**	**(0.88**	–	**1.03)**
High School Diploma	**0.70**	**(0.63**	–	**0.76)**	**0.74**	**(0.68**	–	**0.79)**	**0.85**	**(0.79**	–	**0.92)**	**0.71**	**(0.66**	–	**0.76)**	**0.94**	**(0.87**	–	**1.01)**
≥Bachelor’s Degree	**0.66**	**(0.59**	–	**0.73)**	**0.69**	**(0.63**	–	**0.75)**	**0.78**	**(0.71**	–	**0.85)**	**0.60**	**(0.55**	–	**0.66)**	**0.78**	**(0.71**	–	**0.85)**
**Obesity**																				
Underweight	**0.04**	**(0.03**	–	**0.07)**	**0.27**	**(0.22**	–	**0.33)**	**0.63**	**(0.56**	–	**0.71)**	**0.45**	**(0.38**	–	**0.52)**	**0.45**	**(0.38**	–	**0.54)**
Normal weight	1.00	-		-	1.00	-		-	1.00	-		-	1.00	-		-	1.00	-		-
Overweight	**31.16**	**(29.14**	–	**33.33)**	**2.46**	**(2.35**	–	**2.58)**	**1.86**	**(1.77**	–	**1.96)**	**2.51**	**(2.39**	–	**2.64)**	**2.18**	**(2.08**	–	**2.29)**
**Smoking Status**																				
Non-smoker	1.00	-		-	1.00	-		-	1.00	-		-	1.00	-		-	1.00	-		-
Smoker	**1.13**	**(1.04**	–	**1.22)**	**1.66**	**(1.56**	–	**1.76)**	**1.30**	**(1.22**	–	**1.39)**	**0.96**	**(0.90**	–	**1.02)**	**1.02**	**(0.96**	–	**1.09)**
**Drinking Status**																				
Non-drinker	1.00	-		-	1.00	-		-	1.00	-		-	1.00	-		-	1.00	-		-
Drinker	1.01	(0.95	–	1.08)	1.14	(1.09	–	1.20)	0.61	(0.59	–	0.65)	1.23	(1.17	–	1.30)	1.20	(1.14	–	1.27)
**Stress Status**																				
Low stress	1.00	-		-	1.00	-		-	1.00	-		-	1.00	-		-	1.00	-		-
High stress	1.06	(1.00	–	1.13)	1.07	(1.01	–	1.13)	1.04	(0.98	–	1.09)	1.06	(1.01	–	1.12)	1.04	(0.99	–	1.10)
**Year**																				
2007	1.00	-		-	1.00	-		-	1.00	-		-	1.00	-		-	1.00	-		-
2008	**0.89**	**(0.78**	–	**1.02)**	**0.93**	**(0.83**	–	**1.05)**	**0.24**	**(0.21**	–	**0.26)**	**1.11**	**(0.98**	–	**1.25)**	**1.32**	**(1.17**	–	**1.49)**
2009	**0.71**	**(0.62**	–	**0.81)**	**0.90**	**(0.80**	–	**1.01)**	**0.22**	**(0.20**	–	**0.25)**	**1.66**	**(1.48**	–	**1.87)**	**1.15**	**(1.02**	–	**1.30)**
2010	**0.67**	**(0.58**	–	**0.77)**	**0.90**	**(0.80**	–	**1.01)**	**0.22**	**(0.20**	–	**0.25)**	**1.76**	**(1.56**	–	**1.99)**	**1.09**	**(0.96**	–	**1.23)**
2011	**0.77**	**(0.67**	–	**0.88)**	**0.95**	**(0.84**	–	**1.06)**	**0.19**	**(0.17**	–	**0.21)**	**1.46**	**(1.30**	–	**1.65)**	**1.15**	**(1.02**	–	**1.30)**
2012	**0.66**	**(0.57**	–	**0.76)**	**0.92**	**(0.82**	–	**1.03)**	**0.25**	**(0.23**	–	**0.28)**	**1.42**	**(1.25**	–	**1.61)**	**1.26**	**(1.11**	–	**1.43)**
2013	**0.57**	**(0.49**	–	**0.66)**	**0.91**	**(0.81**	–	**1.03)**	**0.24**	**(0.22**	–	**0.27)**	**1.33**	**(1.17**	–	**1.50)**	**1.45**	**(1.28**	–	**1.65)**
2014	**0.69**	**(0.60**	–	**0.79)**	**0.96**	**(0.85**	–	**1.08)**	**0.23**	**(0.21**	–	**0.26)**	**1.28**	**(1.13**	–	**1.46)**	**1.37**	**(1.20**	–	**1.55)**
2015	0.93	(0.81	–	1.08)	0.95	(0.85	–	1.08)	-				1.41	(1.25	–	1.60)	1.58	(1.40	–	1.79)

* CI: Confidence interval. The bolds here are to show that they are the significant variables.

**Table 5 ijerph-16-04486-t005:** Subgroup analysis.

	Use of Labelling	Metabolic Syndrome			
		OR	95% CI		
			Lower		Upper
**Sex**					
Male	1.00	1.13	0.98	-	1.31
Female	1.00	1.10	0.98	-	1.23
**Age**					
20–29	1.00	1.35	0.95	–	1.91
30–39	1.00	1.16	0.95	–	1.42
40–49	1.00	1.04	0.88	–	1.23
50–59	1.00	1.15	0.96	–	1.37
60–69	1.00	1.21	0.95	–	1.54
70–79	1.00	**1.65**	**1.05**	–	**2.60**
≥80	1.00	1.11	0.13	–	9.74
**Region**					
Urban	1.00	1.11	0.97	–	1.27
Rural	1.00	**1.34**	**1.19**	–	**1.50**
**Income (%)**					
Low	1.00	1.20	0.91	–	1.58
Medium-low	1.00	**1.33**	**1.12**	–	**1.59**
Medium-high	1.00	1.11	0.95	–	1.29
High	1.00	**1.19**	**1.02**	–	**1.39**
**Occupation**					
White Collar	1.00	1.15	1.00	–	1.32
Sales and Services	1.00	1.18	0.97	–	1.44
Blue Collar	1.00	**1.21**	**1.05**	–	**1.40**
**Educational Attainment**					
≤Elementary School	1.00	1.10	0.97	–	1.26
Middle School	1.00	1.12	0.99	–	1.28
High School Diploma	1.00	**1.20**	**1.05**	–	**1.38**
≥Bachelor’s Degree	1.00	0.97	0.71	–	1.31
**Obesity**					
Underweight	1.00	2.06	0.41	–	10.39
Normal weight	1.00	**1.46**	**1.28**	–	**1.68**
Overweight	1.00	1.09	0.97	–	1.23
**Smoking Status**					
Non-smoker	1.00	**1.17**	**1.07**	–	**1.29**
Smoker	1.00	**1.39**	**1.11**	–	**1.75**
**Drinking Status**					
Non-drinker	1.00	**1.23**	**1.10**	–	**1.39**
Drinker	1.00	**1.19**	**1.04**	–	**1.36**
**Stress Status**					
Low stress	1.00	**1.21**	**1.09**	–	**1.34**
High stress	1.00	**1.32**	**1.11**	–	**1.56**

The bolds here are to show that they are the significant variables.

**Table 6 ijerph-16-04486-t006:** Subgroup analysis of metabolic syndrome’s components.

	Use of Labeling	Central Obesity				Use of Labeling	High Triglycerides				Use of Labeling	Low HDL Cholesterol				Use of Labeling	High Blood Pressure				Use of Labeling	High Fasting Plasma Glucose			
		OR	95% CI				OR	95% CI				OR	95% CI				OR	95% CI				OR	95% CI		
			Lower		Upper			Lower		Upper			Lower		Upper			Lower		Upper			Lower		Upper
**Sex**																									
Male	1.00	1.08	0.92	–	1.27	1.00	1.03	0.91	–	1.17	1.00	**1.30**	**1.11**	–	**1.52**	1.00	0.98	0.87	–	1.11	1.00	1.06	0.93	–	1.21
Female	1.00	**1.20**	**1.08**	–	**1.34**	1.00	1.07	0.97	–	1.19	1.00	**1.15**	**1.05**	–	**1.25**	1.00	1.02	0.92	–	1.13	1.00	**1.11**	**1.00**	–	**1.24**
**Age**																									
20–29	1.00	1.30	1.00	–	1.70	1.00	1.20	0.94	–	1.53	1.00	1.10	0.90	–	1.34	1.00	1.09	0.83	–	1.43	1.00	**1.50**	**1.06**	–	**2.13**
30–39	1.00	**1.23**	**1.03**	–	**1.46**	1.00	1.08	0.92	–	1.27	1.00	**1.19**	**1.03**	–	**1.37**	1.00	1.12	0.93	–	1.34	1.00	1.07	0.89	–	1.27
40–49	1.00	1.06	0.89	–	1.26	1.00	1.08	0.92	–	1.26	1.00	**1.21**	**1.04**	–	**1.41**	1.00	0.96	0.82	–	1.11	1.00	1.14	0.98	–	1.33
50–59	1.00	**1.32**	**1.08**	–	**1.60**	1.00	1.09	0.92	–	1.29	1.00	1.15	0.96	–	1.38	1.00	0.98	0.83	–	1.14	1.00	1.06	0.90	–	1.24
60–69	1.00	1.24	0.93	–	1.64	1.00	0.98	0.78	–	1.24	1.00	1.23	0.94	–	1.61	1.00	1.03	0.82	–	1.30	1.00	1.09	0.87	–	1.36
70–79	1.00	1.33	0.76	–	2.33	1.00	1.47	0.89	–	2.41	1.00	1.42	0.87	–	2.31	1.00	0.89	0.56	–	1.42	1.00	1.06	0.70	–	1.59
≥80	1.00	5.56	0.15	–	201.86	-					1.00	1.42	0.11	–	18.79	1.00	0.52	0.05	–	5.27	1.00	0.91	0.12	–	6.78
**Region**																									
Urban	1.00	1.11	0.97	–	1.27	1.00	1.06	0.94	–	1.20	1.00	**1.12**	**1.00**	–	**1.26**	1.00	1.07	0.95	–	1.20	1.00	1.06	0.94	–	1.20
Rural	1.00	**1.32**	**1.18**	–	**1.49**	1.00	**1.22**	**1.10**	–	**1.35**	1.00	**1.28**	**1.16**	–	**1.41**	1.00	**1.14**	**1.03**	–	**1.26**	1.00	**1.18**	**1.06**	–	**1.31**
**Income (%)**																									
Low	1.00	1.28	0.94	–	1.75	1.00	0.99	0.78	–	1.27	1.00	1.20	0.93	–	1.54	1.00	1.15	0.89	–	1.47	1.00	0.97	0.75	–	1.25
Medium-low	1.00	**1.37**	**1.15**	–	**1.64**	1.00	**1.19**	**1.02**	–	**1.40**	1.00	1.15	0.98	–	1.33	1.00	0.98	0.84	–	1.14	1.00	1.11	0.94	–	1.30
Medium-high	1.00	1.09	0.93	–	1.27	1.00	1.09	0.95	–	1.26	1.00	**1.23**	**1.08**	–	**1.41**	1.00	1.11	0.97	–	1.28	1.00	1.09	0.94	–	1.26
High	1.00	**1.20**	**1.03**	–	**1.40**	1.00	**1.16**	**1.02**	–	**1.33**	1.00	**1.17**	**1.03**	–	**1.34**	1.00	1.09	0.95	–	1.25	1.00	**1.17**	**1.02**	–	**1.35**
**Occupation**																									
White Collar	1.00	1.11	0.97	–	1.28	1.00	1.11	0.98	–	1.25	1.00	**1.20**	**1.07**	–	**1.36**	1.00	0.99	0.88	–	1.12	1.00	1.12	0.99	–	1.28
Sales and Services	1.00	1.15	0.93	–	1.43	1.00	**1.22**	**1.02**	–	**1.47**	1.00	1.15	0.94	–	1.41	1.00	1.10	0.92	–	1.30	1.00	0.93	0.78	–	1.11
Blue Collar	1.00	**1.32**	**1.15**	–	**1.52**	1.00	1.08	0.95	–	1.23	1.00	**1.19**	**1.06**	–	**1.34**	1.00	1.10	0.97	–	1.26	1.00	**1.20**	**1.05**	–	**1.38**
**Educational Attainment**																									
≤Elementary School	1.00	1.25	0.87	–	1.80	1.00	0.98	0.73	–	1.31	1.00	1.10	0.80	–	1.51	1.00	0.97	0.73	–	1.30	1.00	0.95	0.71	–	1.26
Middle School	1.00	1.24	0.92	–	1.67	1.00	0.94	0.73	–	1.21	1.00	1.24	0.95	–	1.60	1.00	**0.77**	**0.61**	–	**0.98**	1.00	1.14	0.88	–	1.46
High School Diploma	1.00	**1.21**	**1.05**	–	**1.39**	1.00	**1.25**	**1.10**	–	**1.42**	1.00	**1.14**	**1.01**	–	**1.28**	1.00	1.06	0.94	–	1.21	1.00	**1.19**	**1.05**	–	**1.36**
≥Bachelor’s Degree	1.00	1.13	0.99	–	1.30	1.00	1.01	0.89	–	1.14	1.00	**1.21**	**1.07**	–	**1.36**	1.00	1.00	0.88	–	1.13	1.00	1.03	0.90	–	1.17
**Obesity**																									
Underweight	1.00	1.19	0.32	–	4.51	1.00	0.90	0.47	–	1.69	1.00	1.00	0.70	–	1.43	1.00	1.03	0.60	–	1.77	1.00	1.48	0.74	–	2.94
Normal weight	1.00	**1.34**	**1.19**	–	**1.51**	1.00	**1.26**	**1.13**	–	**1.40**	1.00	**1.23**	**1.12**	–	**1.35**	1.00	**1.15**	**1.03**	–	**1.27**	1.00	**1.17**	**1.05**	–	**1.31**
Overweight	1.00	1.08	0.94	–	1.24	1.00	1.06	0.94	–	1.19	1.00	**1.20**	**1.05**	–	**1.36**	1.00	1.06	0.94	–	1.20	1.00	1.09	0.96	–	1.23
**Smoking Status**																									
Non-smoker	1.00	**1.20**	**1.09**	–	**1.32**	1.00	**1.12**	**1.03**	–	**1.22**	1.00	**1.21**	**1.12**	–	**1.32**	1.00	1.06	0.97	–	1.15	1.00	**1.12**	**1.03**	–	**1.23**
Smoker	1.00	1.25	0.97	–	1.60	1.00	1.16	0.97	–	1.40	1.00	1.06	0.86	–	1.31	1.00	1.18	0.97	–	1.44	1.00	1.12	0.91	–	1.38
**Drinking Status**																									
Non-drinker	1.00	**1.22**	**1.07**	–	**1.40**	1.00	**1.15**	**1.02**	–	**1.30**	1.00	**1.21**	**1.08**	–	**1.35**	1.00	1.10	0.97	–	1.24	1.00	1.07	0.95	–	1.22
Drinker	1.00	**1.23**	**1.09**	–	**1.38**	1.00	**1.12**	**1.01**	–	**1.24**	1.00	**1.20**	**1.08**	–	**1.33**	1.00	1.08	0.97	–	1.19	1.00	**1.15**	**1.04**	–	**1.28**
**Stress Status**																									
Low stress	1.00	**1.23**	**1.11**	–	**1.36**	1.00	**1.11**	**1.01**	–	**1.22**	1.00	**1.27**	**1.16**	–	**1.39**	1.00	1.06	0.97	–	1.16	1.00	**1.11**	**1.01**	–	**1.22**
High stress	1.00	**1.25**	**1.06**	–	**1.48**	1.00	**1.27**	**1.09**	–	**1.48**	1.00	1.06	0.92	–	1.22	1.00	**1.24**	**1.07**	–	**1.45**	1.00	**1.19**	**1.01**	–	**1.39**

The bolds here are to show that they are the significant variables.

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
