# Peer review of "Association between Use of Nutritional Labeling and the Metabolic Syndrome and Its Components"

_ijerph, 2019, doi:10.3390/ijerph16224486_

Round 1

Reviewer 1 Report

Authors of research article entitled "Association between Use of Nutritional Labeling and the Metabolic Syndrome and its components" have presented an interesting finding stating strong association between usage of food labels and metabolic syndrome. In the subgroup analysis, drinking status, smoking status
and stress status were significant contributing factors to metabolic syndrome.

Minor comments:

1.Authors have stated that the presence of MetS was measured using the International Diabetes Federation’s definition for Asians. And at the same time Korean Academy of Medical Sciences definition of MetS has been used. So either one must be followed, though both bodies considerations for defining MetS are more or less the same.

2. Besides nutritional labeling what is equally important is type of nutrition and nutritional break up (with respect to fat:carbohydrate: protein) included in diet. No data has been provided about nutritional value or type in the study. The same should be provided in form of data.

Author Response

1.Authors have stated that the presence of MetS was measured using the International Diabetes Federation’s definition for Asians. And at the same time Korean Academy of Medical Sciences definition of MetS has been used. So either one must be followed, though both bodies considerations for defining MetS are more or less the same.

We are sorry for the major mistake. The definition of the Korean Academy of Medical Sciences was used when processing the data. We revised the manuscript that the guidelines of the Korean Academy of Medical Sciences was used.

2. Besides nutritional labeling what is equally important is type of nutrition and nutritional break up (with respect to fat:carbohydrate: protein) included in diet. No data has been provided about nutritional value or type in the study. The same should be provided in form of data.

We were also aware of the problem that type of nutrition included in the diet is also important when making a discussion about MetS. However in our study we are trying to show a relationship between use of nutritional labeling and MetS. Also, in this manuscript we were working under assumption that the group using nutritional labeling tend to show concern on type of nutrition on their diet. So under that assumption if we show that there is a decrease in MetS in the group using nutritional labeling then we can make a proposal that by educating people to properly use nutritional labeling then we can expect a decrease in MetS.

We see that there was not a enough explanation in the manuscript so we included some explanation about the information above in the manuscript conclusion part.

Reviewer 2 Report

Review of the manuscript entitled, “Association between Use of Nutritional Labeling and the Metabolic Syndrome and its components”

In the present study the authors evaluated a potential association between the diagnosis of metabolic syndrome (MetS) and nutritional label awareness. Out of the overall study population (n = 41,667, Korean population) 27.4% were diagnosed with MetS. Not using nutritional labeling was associated with an increased risk (OR: 1.24, 95% CI 1.14-1.35). Subgroup analysis indicated smoking status, drinking status, and stress status as influencing factors.

This study deals with an important issue from scientific and clinical perspectives. Overall, the paper is nicely presented. Nevertheless, there are several points that should be considered by the authors before a final recommendation can be made.

A clear hypothesis is missing. The authors should state a hypothesis based on literature findings discussed in the intro section. The selection process of the analysed population out of the overall KNHANES data should be described in more detail. Nutritional labeling is used as a dichotomous variable. Are you sure that people can be strictly divided in those using and those not using labeling? What about sometimes using labeling? Please, comment on that? The model used for logistic regression analysis should be described in more detail. Which predictor variables have been (why, how) included in the multivariable model? The percentages shown in table 1 are a bit misleading, at least for me. Numbers of yes and no should be 100% within subgroups instead of summarizing numbers within lines. Please explain. Table 1 cont. and Table 4 cont. should better be presented as extra tables. What means an OR > 999.999? I do not see “drinking status” as a significant factor?! Please explain. The novelty of the presented findings should be better highlighted (intro and discussion section). Proper proof reading is necessary.

Author Response

1. A clear hypothesis is missing. The authors should state a hypothesis based on literature findings discussed in the intro section.

We included a clear hypothesis that we tried to show in the introduction section.

2. The selection process of the analysed population out of the overall KNHANES data should be described in more detail.

Thank you for your kind suggestion. We have added the following information regarding the KNHANES to our manuscript:

The target population of KNHANES comprises non-institutionalized Korean citizens residing in Korea. The sampling plan follows a multi-stage clustered probability design. For example, in the 2011 survey, 192 primary sampling units (PSUs) were drawn from approximately 200 000 geographically defined PSUs for the whole country.2 A PSU consisted of an average of 60 households, and 20 final target households were sampled for each PSU using systematic sampling; in the selected households, individuals aged 1 year and over were targeted. The number of participants is shown in Table 1. The numbers of participants of the first three surveys (1998, 2001 and 2005) were approximately 35 000 in each survey. From 2007 the survey became a continuous programme with about 10 000 individuals each year except for the year 2007, when the number of participants was half of that of other years as the 2007 survey was conducted during a half-year (from July through December). All statistics of this survey have been calculated using sample weights assigned to sample participants (Oh et al., 2014).

3. Nutritional labeling is used as a dichotomous variable. Are you sure that people can be strictly divided in those using and those not using labeling? What about sometimes using labeling? Please, comment on that?

We were also aware of the problem that Nutritional labeling usage is hard to be described as a dichotomous variable. But the KNHANES data we used described the usage of nutritional labeling as a dichotomous variable and we had to use the data. This is the limit of our study and further studies should be needed for the people sometimes using labeling.

4. The model used for logistic regression analysis should be described in more detail. Which predictor variables have been (why, how) included in the multivariable model?

The factors we used are profound factors for MetS and as we wanted to see the relationship between use of nutritional labeling and MetS we had to rule out their influences on MetS. So the 10 variables, Sex, Age, Region, Household Income, Occupation, Educational Attainment, Obesity, Smoking Status, Drinking Status, Stress Status were chosen.

5. The percentages shown in table 1 are a bit misleading, at least for me. Numbers of yes and no should be 100% within subgroups instead of summarizing numbers within lines. Please explain.

Sorry for the confusion that this may have caused. The % in our table should be read horizontally so that the number of individuals who have Metabolic Syndrome or don’t have Metabolic Syndrome add up to 100%. We decided to make the numbers within lines add up to 100% because we thought that the proportion of individuals in each category would be more important to show rather than the proportion of individuals within each subgroup. Please inform us if you disagree with our perspective and we will change all % to be within subgroups instead of between lines.

6. Table 1 cont. and Table 4 cont. should better be presented as extra tables. What means an OR > 999.999?

Sorry for the confusion that this error would have caused the reviewer. There were too few people I this subgroup which is why the statistics came up like this. We have deleted this part of our manuscript so that it no longer shows this error. Thank you for your insight.

7. I do not see “drinking status” as a significant factor?! Please explain.

Sorry for the confusion we have caused. We were talking about subgroup analysis in the section. When we look at table 4 we can see that groups which drink are more vulnerable to MetS when not using Nutritional Labeling. We see that they are not a significant factor in table 2 but here we were talking about table 4 and they are a significant factor.

8. The novelty of the presented findings should be better highlighted (intro and discussion section).

We see that there weren’t enough explanation about the novelty of our work in the introduction section. We added some more information about the novelty of the presented findings in the intro section and also the conclusion section.

Round 2

Reviewer 2 Report

The authors responded adequately to most of my concerns/comments. Some weaknesses remain due to data available. I my opinion, table 1 cont. and table 4 cont. should be presented as separate tables, e.g. table 2, with own legends. Your opinion?

No further comments.

Author Response

Thank you for your insight. We also think that table 1 cont. and table 4 cont. should be presented as separate tables. We presented the tables in separate tables and the corresponding body contents were changed.

Thank you again for your deep insight.